# Analysis of High-Friction Surface Texture with Respect to Friction and Wear

**Cibi Pranav [1,*], Minh-Tan Do [2] and Yi-Chang Tsai [1]**

[1] Georgia Institute of Technology, Atlanta, GA 30332, USA; james.tsai@ce.gatech.edu
[2] AME-EASE, University Gustave Eiffel, 44344 Bouguenais, France; minh-tan.do@univ-eiffel.fr
* Correspondence: cibipranav@gatech.edu

**Abstract:** High Friction Surfaces (HFS) are applied to increase friction capacity on critical roadway sections, such as horizontal curves. HFS friction deterioration on these sections is a safety concern. This study deals with characterization of the aggregate loss, one of the main failure mechanisms of HFS, using texture parameters to study its relationship with friction. Tests are conducted on selected HFS spots with different aggregate loss severity levels at the National Center for Asphalt Technology (NCAT) Test Track. Friction tests are performed using a Dynamic Friction Tester (DFT). The surface texture is measured by means of a high-resolution 3D pavement scanning system (0.025 mm vertical resolution). Texture data are processed and analyzed by means of the MountainsMap software. The correlations between the DFT friction coefficient and the texture parameters confirm the impact of change in aggregates' characteristics (including height, shape, and material volume) on friction. A novel approach to detect the HFS friction coefficient transition based on aggregate loss, inspired by previous works on the tribology of coatings, is proposed. Using the proposed approach, preliminary outcomes show it is possible to observe the rapid friction coefficient transition, similar to observations at NCAT. Perspectives for future research are presented and discussed.

**Keywords:** high friction surface; aggregates loss; surface texture; coating

## 1. Introduction

The High Friction Surface (HFS) has been used as one of the low-cost safety counter-measures to address high friction demand concerns on curved roadways. It was developed in the 1960s in the UK by the Greater London Council and the Transport and Road Research Laboratory to restore friction on high-crash road sections [1]. The technology has been adopted in many countries like New-Zealand [2] and the United States [3]. HFS is applied as a thin coating (compared to the thickness of a pavement) on an existing road surface. It is composed of a layer of wear-resistant aggregate like calcined bauxite bonded to the pavement surface with polymer resin binder. Thanks to the high quality and the small size (3–4 mm) of the aggregates, HFS increases the pavement friction value immediately and has proven to significantly reduce roadway departure crashes and wet pavement crashes on horizontal curves and ramps [3]. Figure 1 shows a cross section of pavement installed with the HFS cored out, and it shows the typical dimensions and illustration of HFS.

Properly constructed HFS on pavements in good condition typically maintains a high friction value throughout its expected life (which varies from seven to twelve years depending on the construction quality, roadway geometry, and traffic and environmental conditions). It has been proved that the benefits, in terms of crash reduction, far exceed the cost of construction if the surfacing lasts seven years or more [1]. However, HFS can fail prematurely (before seven years) in the form of delamination failure, aggregate loss, or cohesion failure [1] and can, potentially, reduce friction capacity.

Since HFS is applied at critical operation locations (such as horizontal curves) with high friction demand, friction reductions are a serious safety concern, and HFS friction deterioration needs to be detected in its early stages. To detect friction deterioration

in its early stages, continuous and long-term friction monitoring is essential. However, some of the existing methods of friction measurement used by transportation agencies are not sustainable for network-wide, long-term friction monitoring (e.g., friction measured with a Dynamic Friction Tester is traffic-disrupting, time-consuming, expensive, and cumbersome). Moreover, based on field observation at the National Center for Asphalt Technology (discussed in the HFS friction loss section below), there is no clear long-term friction behavior trends to predict the occurrence of rapid friction loss.

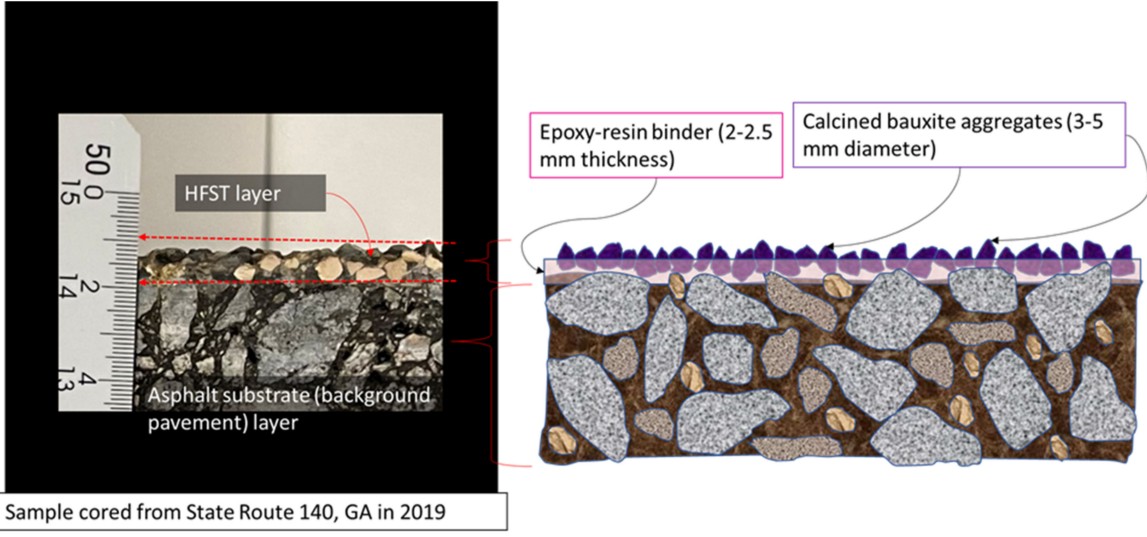

**Figure 1.** Cross section of HFS and illustration showing the typical dimensions of epoxy and calcined bauxite.

Therefore, there is an urgent need to investigate other methods to characterize the HFS and explore alternate performance measures (e.g., aggregate loss, which is one of the major failures associated with HFS friction loss) to monitor HFS friction and predict its failure before irreversible and rapid friction deterioration occurs. Based on the well-known link between friction and surface texture, it was thought that a texture-based characterization of HFS aggregate loss would improve understanding of the mechanisms involved in HFS aggregate loss process and predict HFST friction deterioration before damages are visible.

The objective of this paper is, thus, to characterize different stages of aggregate loss using texture parameters, study its relationship with HFS friction using correlation analysis, and interpret the aggregate loss and friction deterioration behavior.

The paper is organized as follows. After the present introduction, the background section discusses the engineering problem related to HFS friction loss and relevant studies that characterized HFS texture; the subsequent section describes the texture parameter categories, experiments conducted at the National Center for Asphalt Technology (NCAT) Test Track; then, the employed texture parameters and their correlation with friction are presented; next, the potential use of texture analysis to understand damage mechanisms is discussed; finally, conclusions are drawn and recommendations for future investigations are made.

## 2. Background

### 2.1. HFS Friction Loss

Figure 2 shows the plot of long-term friction performance (measured as a Skid Number) of HFS installed in 2005 at the NCAT test track. The friction performance was measured with a skid trailer fitted with a ribbed tire and run at about 60 km/h (40 mph). It is observed that there is no clear deterioration trend based on Skid Number readings for close to nine years. A rapid friction drop is also observed after nine years in 2014.

Consequently, this situation causes significant safety concern because transportation agencies cannot reliably predict the friction deterioration and plan for timely actions to miti-

gate the problem, such as performing a detailed survey, setting up warning signs, replacing pavement, resurfacing pavement, etc. Although, this is not a generalized observation in real-world roadway conditions, the few available data, such as those collected at the NCAT test track, are valuable because they confirm the need to better understand the transition from mild friction decrease (from 2007 to 2014 in Figure 2) to rapid friction decrease within a few months. The transition from a mild evolution of friction deterioration to a rapid deterioration is observable on test tracks, especially at NCAT where the testing can be performed under accelerated pavement testing conditions (more than 50,000 axle passes of load, i.e., 100,000 equivalent single axle loads, every week). Such extremely deteriorated real-world in-service HFST sites are usually replaced by road authorities however, the road authorities also need to take proactive safety and maintenance actions before HFS sections are extremely damaged because it is tied to friction reduction.

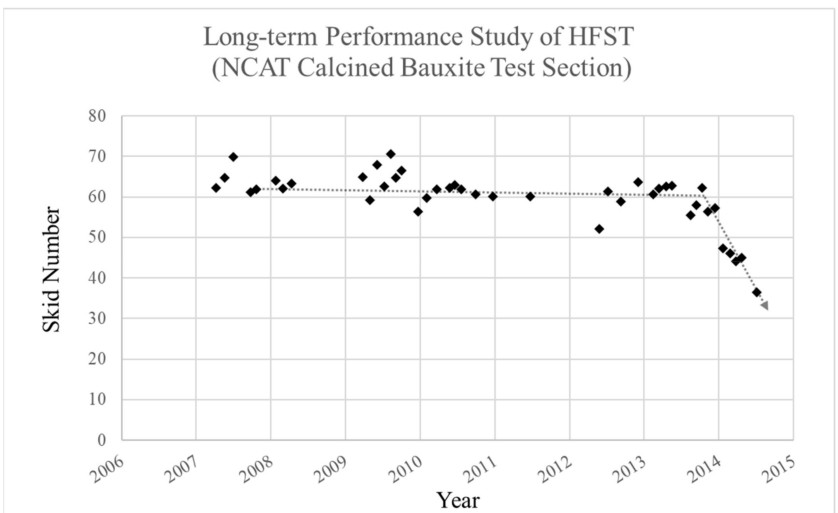

**Figure 2.** Evolution of friction of a HFS surface. Data courtesy of NCAT.

A long-term, real-world friction study by Anderson et al. [4] showed that the friction values (skid number) were exceptionally high for the Tyregrip® (an HFST system involving calcined bauxite and epoxy-resin binder), above the 70s, and were maintained throughout the five-year evaluation period of the study; the study included a 17,000 annual average daily traffic (AADT) ramp in Washington state. Unlike the authors in [4], who continuously studied single HFST sites over five years, Scully [5] analyzed Kentucky's HFST friction performance at the network level using different HFST sites to present his findings. Scully noted that the calcined bauxite HFST friction values were high and stable for five years, based on the DFT friction measured in 2015 and based on several sites of different ages in Kentucky.

Heitzman et al. [6] studied the friction performance of eight aggregates, including granite, calcined bauxite, flint, basalt, silica, slag, emery, and taconite, applied on top of epoxy-resin binder at the NCAT test track. The test sections were subject to approximately 2.6 million ESALs (equivalent to 350,000 18-wheel tractor-trailer units) in six months. Granite, calcined bauxite, and flint aggregate sections were exposed up to 10 million ESALs, corresponding to 24 months of accelerated truck traffic conditioning. The dynamic friction tester friction (measured at 40 mph) and the locked-wheel skid tester SN values showed initial reduction and stabilized. It was found that the calcined bauxite section maintained higher friction levels (more than 0.75 DFT40 friction) compared to rest of the aggregates (less than 0.60 DFT40 friction) throughout the testing period.

Wear tests conducted in the laboratory on HFS specimens using different types of aggregates (as reported by Woodward and Friel [1]) using the Road Test Machine at Ulster University, show a rapid decrease of friction (measured with the British Pendulum Tester) with the number of wheel passes (full-size tires with an inflation pressure of 0.2 MPa)

followed by a mild decrease. However, the overall BPT skid resistance value stayed above 75 during the period of mild decrease.

The main differences between the evolutions observed in Figure 2 and references [1,6] are (a) test duration, which is more than nine years in Figure 2, and (b) severity of the simulated traffic, which would be less in [1] in terms of applied loads.

### 2.2. HFS Failure Mechanisms

In a comparison of experiences in New Zealand and overseas, Waters identified three main failure mechanisms of HFS [2]:

- Delamination failure: the whole HFS layer (binder and aggregates) is detached from the substrate.
- Aggregate loss: the aggregates are detached (sometime with the binder) from the layer.
- Cohesion failure: the HFS layer is detached with part of the substrate.

Bennert et al. [7] defined six failure mechanisms, which include those cited in [2] with some variants related to the crack patterns. All authors agree on the causes of the failures: delamination is generally linked to the installation and the curing of the binder, and aggregate loss (resp. cohesion failure) is linked to the shear stress at the aggregate-binder (resp. HFS-substrate) interface, which is very high because of the high friction demand. In the present paper, the failure mechanism of interest is the aggregate loss because it represents the predominant failure mode at the test site (see the experiment set-up and data collection). Discussions in the rest of the text will be focused on this mechanism (readers can refer, for example, to works in [1] and [7] for more details on the other mechanisms).

In [1,2], it was mentioned that calcined bauxite aggregates have sharp edges, which, together with their small size, produce very high contact pressure with vehicles' tires. Therefore, it can be expected that the stress at the aggregate-binder interface is high, as well. As the epoxy resin binder used in HFS becomes potentially brittle over time, due to physical/chemical aging [8] and/or mechanical deformation [9], it loses its capacity to strongly bind the aggregates. Then, the shear stress at the tire-pavement interface forces the aggregates to be dislodged from the brittle epoxy.

Based on well-known mechanism of friction, known as hysteresis, the friction available to the tires depends on the area of the tire that is deformed by the rough, gripping surfaces in the pavement. In HFS, the sharp calcined bauxite aggregates contribute to deformation of the tire at multiple points. However, HFS aggregate loss results in a reduced number of aggregates to create deformations on the tire, thereby potentially reducing the friction capacity. Moreover, with significant aggregate loss, the tire meets the exposed, smooth, glossy polymer resin layer (epoxy binder) and the underlying pavement surface, which may not provide sufficient grip for the tires and result in a slipperier pavement condition (with low friction).

### 2.3. Characterization of HFS

Based on the above explanation, tests have been developed in the laboratory to characterize HFS in view of predicting performance before its application. Bennert in [7] mentioned two types of tests employed in the United States to assess the bonding capacity of HFS and the durability of the binder. The pull-out test helps to see if the failure occurs in the substrate or the coating, or at the interface of the coating/substrate. Durability tests, by simulating natural weathering (sun radiation and precipitation), allow comparison of rheological properties of new and aged binders and assessment of the hardening effect.

Woodward [1] described a wear test using a rotating table equipped with tires (195/70R14) rolling on the test surfaces to simulate a traffic flow of light vehicles. Tests are stopped at a predefined number of rotations, and the skid resistance of the test surfaces is characterized in terms of friction (using a British Pendulum [10]) and texture (using the volumetric test method [11]). The British Pendulum provides the coefficient of friction between a rubber pad sliding (sliding speed of around 12 km/h) on a test surface previously wetted. The volumetric test provides a so-called mean texture depth (MTD) by dividing

a fixed volume of fine glass beads by the diameter of the circular patch they form; this parameter is used to characterize the drainage capacity of the road surface, which concerns the macrotexture (surface irregularities whose dimensions range between 0.1 mm and 20 mm vertically, and between 0.5 mm and 50 mm horizontally [12]).

Other authors [13–17] have exploited topographical 2D and 3D maps, measured by laser or other optical noncontact methods, to provide more surface characteristics than the MTD. Mean profile depth (MPD) is the most used texture parameter to characterize HFS surface [13–15]. Some authors [16,17] also use the so-called ridge-to-valley depth (RVD). Figure 3 provides an illustration of MPD and RVD parameters:

- Mean profile depth is defined by the following formula [12]:

$$MPD = \text{Mean peak level} - \text{average level}$$

where Mean peak level is the average of the two peak levels (numbered one and two, respectively, in Figure 3) determined respectively in the two halves of a profile baseline; and average level is the profile mean level.

Standard ISO 13473-1 [12] provides an empirical formula to relate MTD, as used in [1], to MPD:

$$MTD = 0.2 + 0.8MPD.$$

- Ridge-to-valley depth characterizes the height difference between the highest peak and deepest valley in a profile (Figure 3).

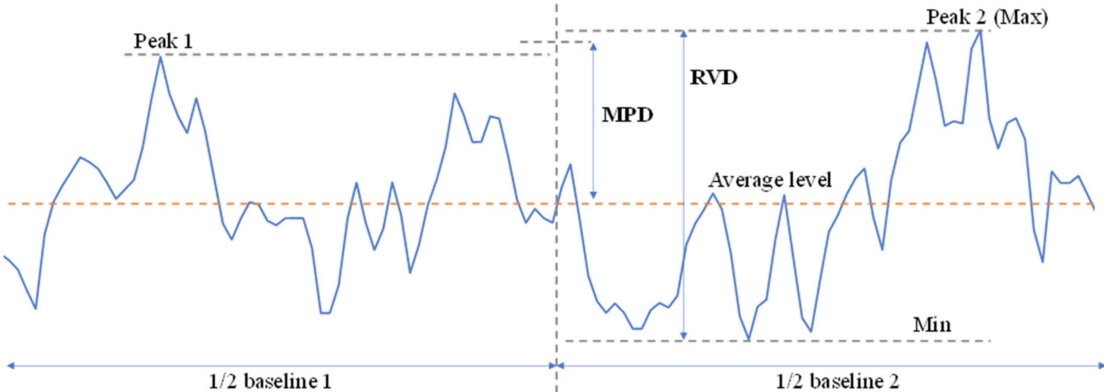

**Figure 3.** Definition of the mean profile depth (MPD) and the ridge-to-valley depth (RVD).

Based on the experiments conducted by Pranav and Tsai [17] using dynamic friction tester, the aggregate loss is strongly correlated to coefficient of friction. Aggregate loss was measured as a fraction of the surface of the HFS coating that is not covered by aggregates under the DFT measurement footprint. As the fraction of aggregate loss increased, the friction decreased linearly, indicating aggregate loss is a major contributing factor to friction deterioration. These authors further showed that MPD and RVD texture parameters can, potentially, differentiate the HFS pavement surface covered with bauxite and HFS pavement surface with large percentage of bauxite aggregate lost.

The self-affine nature of road surfaces was first stated by Persson [18]. Hunnekens [19] has shown that:

$$C(q) \approx q^{-2(H+1)}$$

where C(q) is the power spectral density; q = 1/λ and λ is the length scale or wavelength; H = 3 − D is called the Hurst exponent; and D is the fractal dimension of the surface. Alhasan et. al. [20] consider the multi-scale nature of road surfaces (not limited to the macro- and microtexture scales) and used the Hurst exponent to characterize the HFS pavement. The authors used the Hurst exponent for predicting the HFS friction using the Persson's model.

However, their research did not involve diverse cases of HFS containing different severities of aggregate loss.

Overall, HFS has been characterized using texture parameters, such as MPD, MTD, fractal dimension, and its friction property. However, HFS aggregate loss phenomenon has not been studied previously. Therefore, there are no studies on how the texture parameters and friction property change with aggregate loss. Besides, traditional texture parameters like MPD and MTD are only height parameters and cannot fully describe/characterize the HFST surface topography. Therefore, texture parameters extracted from topographical maps would provide more insight into the characteristics of surface asperities (including asperities' height, shape, distribution, and surface volume) and would be promising to (a) better characterize HFS surface with aggregate loss, and (b) understand the HFST aggregate loss process. Further, the changes in these aggregate loss texture parameters can be used to characterize the aggregate loss relationship with friction and interpret the friction deterioration behavior.

### 2.4. Texture Parameter Categorization

There are many texture parameters, as illustrated by [21]; there is a need to select only parameters that are related to the aggregate loss process and the resulting coefficient of friction. For this purpose, the aggregate-loss failure mode, as described in the background section, is illustrated in Figure 4. Based on the possibility of visually distinguishing surface damages, three deterioration levels are defined (more details in the section "Experimental Set-Up and Data Collection"): minimal, moderate, and severe aggregate loss. For each level, dislocation of aggregates and change in binder are presented.

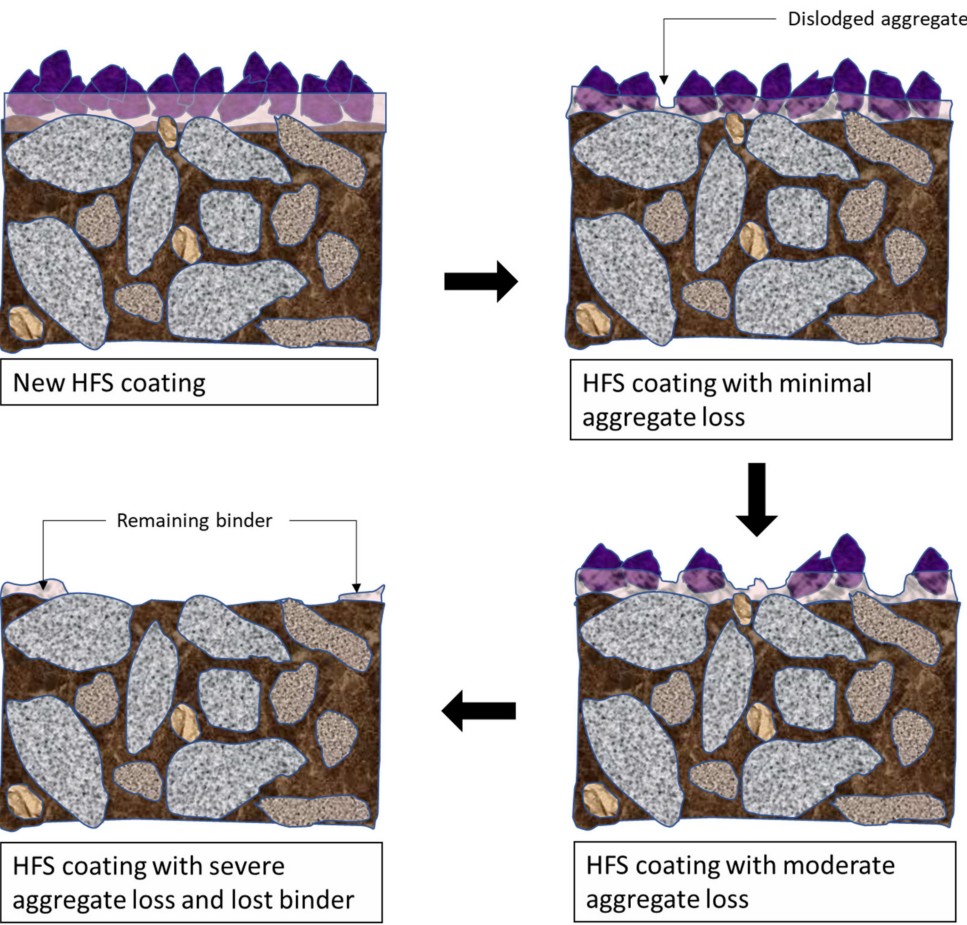

**Figure 4.** Illustration of the aggregate loss process in a HFS coating.

Illustration of the process in Figure 4 helps to identify three categories of texture parameters:

- Parameters related to the aggregate characteristics, e.g., height and shape. This characterization is necessary because the tire traction is generated by the deformation of the tire rubber by the aggregate.
- Parameters related to the distribution of the aggregates, e.g., density and bearing area. It is clear from the illustration that the aggregate loss induces significant changes in the number of points of contact with the tires.
- Parameters related to the aggregate loss physical process, e.g., volume. Because of aggregate detachment, it seems that material and void volumes should be affected.

The following section presents the experimental set up designed to study the texture-friction correlation and includes test location, data collection devices, and data collection procedure. To address the challenges in the previous studies related to multi-parametric nature of friction (i.e., different testing conditions leading to different friction measured at same location), this study utilizes constant friction measurement conditions at the same site and the same time to measure friction but with varying surfaces that have different aggregate loss severities.

### 2.5. Experiment Set-Up and Data Collection

#### 2.5.1. Test Location

The test location is situated in National Center for Asphalt Technology (NCAT), Alabama, USA. The NCAT Test Track study area involved a thin overlay sections of 30 m in length consisting of HFST (calcined bauxite aggregate) embedded in thin epoxy resin layer, installed in 2011. This section is exposed to 100,000 Equivalent Single Axle Loads (ESAL) every week. Figure 5 shows the test location in NCAT Test Track.

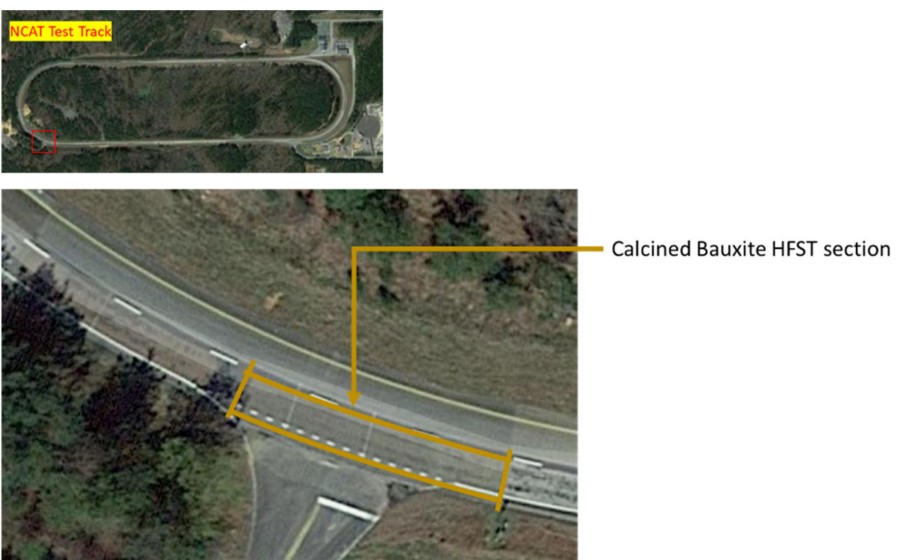

**Figure 5.** NCAT test track (top) and location of HFST section on the west end of the test track (bottom).

#### 2.5.2. Data Collection Devices

Friction tests are performed by means of the Dynamic Friction Tester (DFT). The machine is composed of a measuring unit (Figure 6, upper left) and a control unit. The measuring unit consists of a horizontal fly wheel and disc, which are driven by a motor (Figure 6, upper right). Three rubber sliders (Figure 6, lower right) are attached to the disc by leaf springs. The rubber compound has a Shore hardness of 58 ± 2. The sliders are pressed on the test surface by the weight of the device and are loaded to 11.8 N each. When the disc rotates, the sliders follow a circle of 284 mm in diameter. Friction forces are

generated by the contact between the sliders' edges and the test surface. A water tank (not shown in Figure 6) is used to wet the test surface. The flow is maintained at 3.6 l/min to provide a water depth of 1 mm on the test surface.

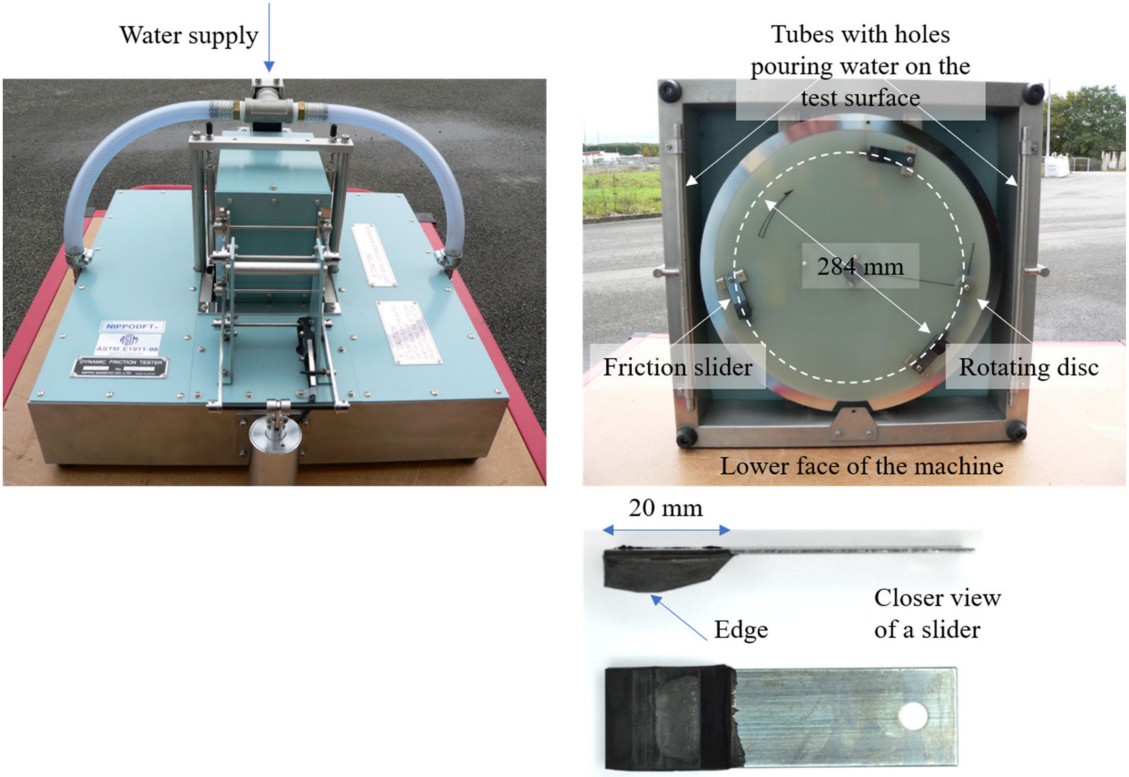

**Figure 6.** Dynamic Friction Tester (DFT) machine.

The test protocol starts with the rotation of the disc and the flow of the water supply. When the tangential speed of the disc reaches a speed of 80 km/h, water flow is closed, and the disc is lowered until the rubber sliders touch the test surface. A braking curve is recorded from 80 km/h until complete stop. In the research study, friction measurement recorded at 60 km/h is used for all the analysis and will be denoted as DFT60 (60 km/h is the usual speed of monitoring devices deployed on the road network).

The surface texture is measured by means of the LS-40 device (also known as Laser Texture Scanner 9500), which is similar to ones developed by AMES Engineering (Figure 7). Characteristics of the device: horizontal resolution 0.05 mm and vertical resolution 0.01 mm). The LS-40 uses a line laser to scan an area of 100 mm by 100 mm and collects both elevation height data and scan intensity image. The protocol is simple (push of a button) and rapid (90 s for s full scan). Data are exported as CSV files and can be analyzed by other software, such as MountainsMap (used in this paper).

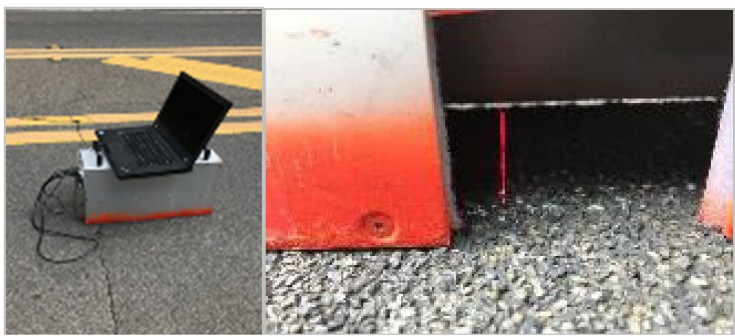

**Figure 7.** Laser scanner (LS-40) machine.

For visual observations of the surface damage, close up images of the surface were collected using a Nikon ShuttlePix P-400R (Figure 8), a high-resolution microscopic imaging camera. It covers an area of 13 mm × 11 mm; it has a resolution of 1600 × 1200 pixels and a horizontal interval 0.009 mm.

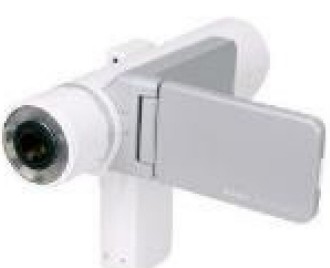 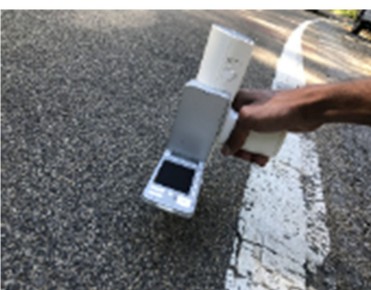

**Figure 8.** ShuttlePix microscopic imaging camera.

2.5.3. Data Collection Procedure

Step 1: Several spots with uniform surface appearance and roughness representing three severity types of HFST aggregate loss were identified manually and marked (Figure 9a). Three severity types were defined as follows:

- Minimal aggregate loss (e.g., 0–20% loss): covered with bauxite aggregate; background surface not visible, high bauxite aggregate density.
- Moderate aggregate loss (e.g., 20–60% loss): majority of surface covered with bauxite aggregate; background surface partially visible, medium bauxite aggregate density.
- Severe aggregate loss (e.g., more than 60% loss): background asphalt aggregate predominately visible, severe bauxite aggregate loss,

Step 2: Three representative spots in each severity level were selected based on homogeneous aggregate loss appearance, resulting in nine spots.

Step 3: 3D surface data was collected using a high-resolution LS-40 surface area scanner; friction was measured using a DFT (one run); and photos were taken using a smart phone and a ShuttlePix microscopic imaging on the selected spots. The footprint of each measurement device is shown in Figure 9b.

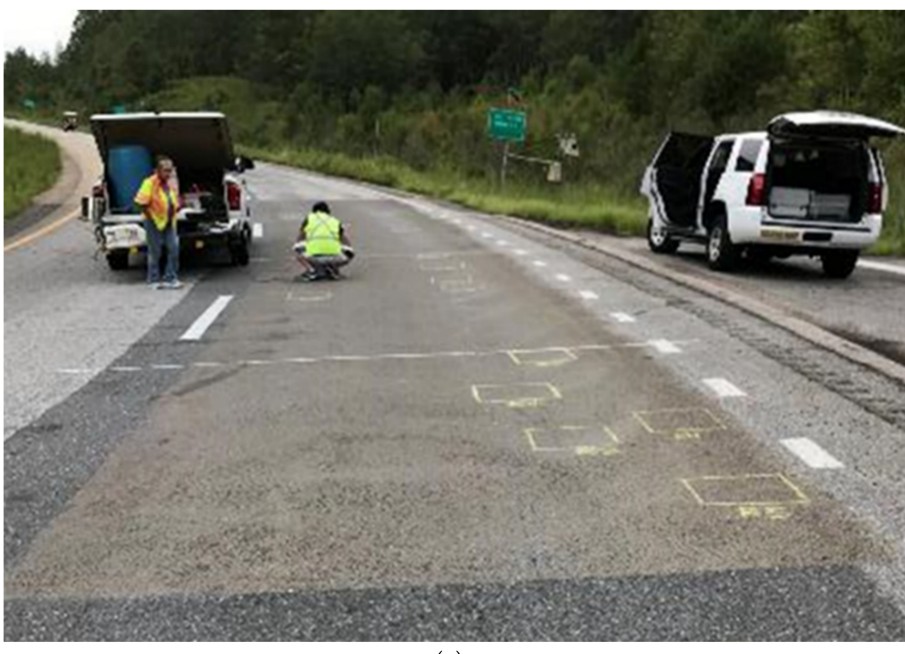

(**a**)

**Figure 9.** *Cont.*

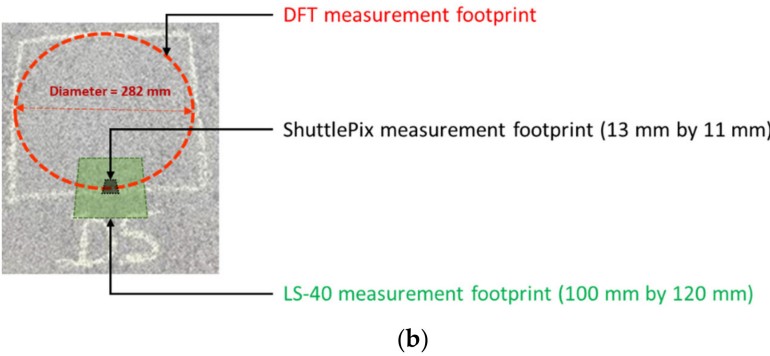

(**b**)

**Figure 9.** (**a**) Manual marking of different aggregate loss severity spots on the HFS test section and (**b**) illustration of different devices' measurement footprint in each spot.

### 2.6. Texture Analysis

2.6.1. Pre-Processing

Topographical data were analyzed by means of the MountainsMap software. Data were first processed using the following procedure:

- Replacing unmeasured points by interpolating values of neighboring points.
- Removing extreme values (0.025% at each end of the height distribution).
- Leveling the maps using the least-square method (regression plane).

As values of the texture parameters should reflect the part of the surface that is in contact with the rubber slider, it was decided to apply a morphological filter to obtain an envelope of the surface (in Figure 10, a profile is shown to facilitate the visualization):

- A dilation operation is first applied by "rolling" a disk over the surface; it creates a dilation line.
- An erosion operation is then applied by rolling the same disk below the dilation line; it creates the upper envelop of the surface.

A diameter of 2 mm was chosen for the rolling disk to simulate the geometry of the edge of the rubber slider (Figure 6).

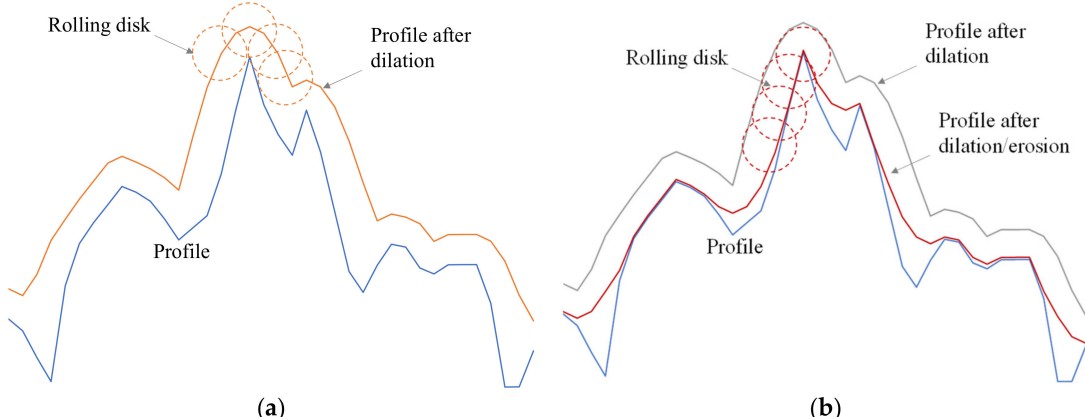

(**a**)　　　　　　　　　　　　　　　　　　　　(**b**)

**Figure 10.** Operations to create an envelope (**a**) dilation (blue: profile; orange: dilation line); (**b**) erosion (blue: profile; grey: dilation line; red: envelop).

2.6.2. Texture Parameters Computation

From the processed data, the following texture parameters were computed based on the three categories defined in the 'texture parameter categorization' section:

- Root mean square height ($S_q$), mean quadratic slope ($S_{dq}$), curvature ($S_{sc}$).
- Density ($S_{ds}$) and the projected area.

- Volume parameters ($V_{mp}$, $V_{mc}$, $V_{vc}$, $V_{vv}$).

The formulae for the texture parameters, as described in [21], are presented below.

(1)   Root mean square height, to characterize the height deviation in the surface.

$$S_q = \sqrt{\frac{1}{MN} \sum_{j=1}^{N} \sum_{i=1}^{M} z^2(x_i, y_j)}$$

(2)   Mean quadratic slope, to characterize the steepness of the asperities in the surface.

$$S_{dq} = \sqrt{\frac{1}{(M-1)(N-1)} \sum_{j=1}^{N} \sum_{i=1}^{M} \left[ \left( \frac{z(x_i, y_j) - z(x_{i-1}, y_j)}{\Delta x} \right)^2 + \left( \frac{z(x_i, y_j) - z(x_i, y_{j-1})}{\Delta y} \right)^2 \right]}$$

(3)   Curvature (at a summit located at xp and yq), to characterize the curvature of the asperity peaks.

$$S_{sc} = -\frac{1}{2} \Delta \frac{1}{n} \sum_{k=1}^{n} \left( \frac{z(x_{p+1}, y_q) + z(x_{p-1}, y_q) - 2z(x_p, y_q)}{\Delta x^2} + \frac{z(x_p, y_{q+1}) + z(x_p, y_{q-1}) - 2z(x_p, y_q)}{\Delta y^2} \right)$$

(4)   Density, to characterize the number of asperities in the surface.

$$S_{ds} = \frac{number\ of\ summits}{(M-1)(N-1)\Delta x \Delta y}$$

where $M$ (resp. $N$) is the number of points measured in the $X$ (resp. $Y$) direction; $\Delta x$ (resp. $\Delta y$) is the sampling interval in the $X$ (resp. $Y$) direction; $z(x, y)$ is the measured height at the point $(x, y)$; and $X_{si}$ is the profile asperity width (figure 10 of [22]).

Figure 11 defines the volume parameters (a) and the projected area (b). The projected area (fraction in red) is obtained by looking at the intersection between a horizontal plane located at a predefined depth in the topographical map. Based on the illustrations presented in Figure 4, it was thought that the projected area will reflect the aggregate loss process. The intersection depth was defined at 1 mm from the highest peak of the topographical map. This value, which should represent the indentation depth of the tire, was chosen arbitrarily; a complete modeling of the envelopment of a road surface by a tire would provide a more precise value, but such a study is out of the scope of the paper. Based on Figure 4, when minimal aggregate is lost, the horizontal plane intersects the sharp and pointed aggregates resulting in lower projected area. At severe aggregate loss, the topmost point is either the exposed substrate aggregate or the remaining epoxy and the projected area is likely to be relatively larger. This is because epoxy and asphalt surface contact area are much flatter and wider than the bauxite aggregate, and they also have low height compared to embedded bauxite.

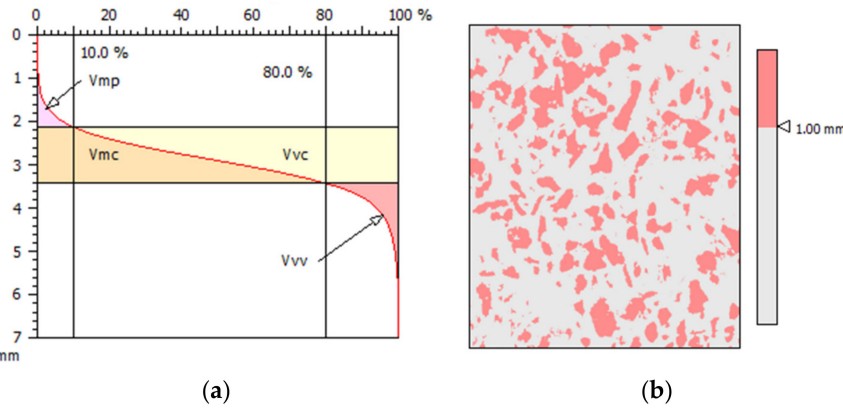

**Figure 11.** (**a**) Abbott curve and the volume parameters; (**b**) projected area (data from MoutainsMap®software).

## 3. Results

### 3.1. Visual Observations

An example for each of the three aggregate loss severity types is shown in Figure 12. In addition, the microscopic image of each severity type is shown in parallel to emphasize the difference and the characteristic's appearance is highlighted. The photos of each spot are enhanced with 400% saturation to show the difference between the calcined bauxite aggregates and the exposed epoxy and background pavement surface.

As all photos were taken nine years after the installation of the coating, and the aging of the epoxy binder is visible (alligator-skin appearance). For the minimal- and moderate-type loss, the calcined bauxite inclusions are still visible; for the severe-type loss, no calcined bauxite aggregate remains, and only the binder and the underlying pavement can be seen.

From topographical maps, isometric views of three surfaces corresponding to the minimal-, moderate- and severe-aggregate loss are shown in Figure 13. The values indicated at the left upper corner are the coefficient of friction measured on these surfaces. A closer view of the asperities (materialized by the rectangle defined by the white dotted line) is shown on the right side of the maps. The same height scale has been applied to all maps to facilitate visual comparisons.

It can be seen that the surface presenting a minimal aggregate loss has higher (more "red" asperities) and sharper asperities (from the close view). On the surface presenting a moderate aggregate loss (coefficient of friction of 0.56), there are relatively fewer higher and sharper asperities compared to the surface with minimal aggregate loss. For the surface presenting a severe aggregate loss (coefficient of friction of 0.35), the asperities are lower than those of the two previous surfaces.

The projected areas of the same topographical maps are shown in Figure 14. The projected area increases as the coefficient of friction decreases. Compared with the aggregate loss process illustrated in Figure 4, this result is partially reasonable. We can see more tire contact with aggregate in minimal and moderate aggregate loss conditions and more contact with binder in severe aggregate loss conditions. On the one hand, the contact points between the tire and the high-friction surface come from the calcined bauxite aggregates with sharp edges (as reported in [1,2]), and so the projected area is low in minimal and moderate aggregate loss. On the other hand, the contact points between the tire and the low-friction surface come mainly from the binder and the underlying pavement (with more rounded aggregates), and so the projected area is high in severe aggregate loss. However, one would expect that the number of contact points (and the projected area) would decrease as the coating is deteriorated from minimal aggregate loss to moderate aggregate loss. This discrepancy needs to be further investigated; it could be possible that the minimal aggregate loss spots chosen in this study may still have excess aggregates on top of the embedded aggregates, leading to much higher elevation of the topmost aggregates. Thus,

only a few points are intersected by the horizontal plane 1mm below the topmost point. After the excess aggregates are removed, it is possible that the projected area decreases with increasing aggregate loss, and, when severe aggregate loss is reached, the projected area suddenly increases; the highest point is now the remaining epoxy, or the background substrate aggregates instead of the calcined bauxite aggregates.

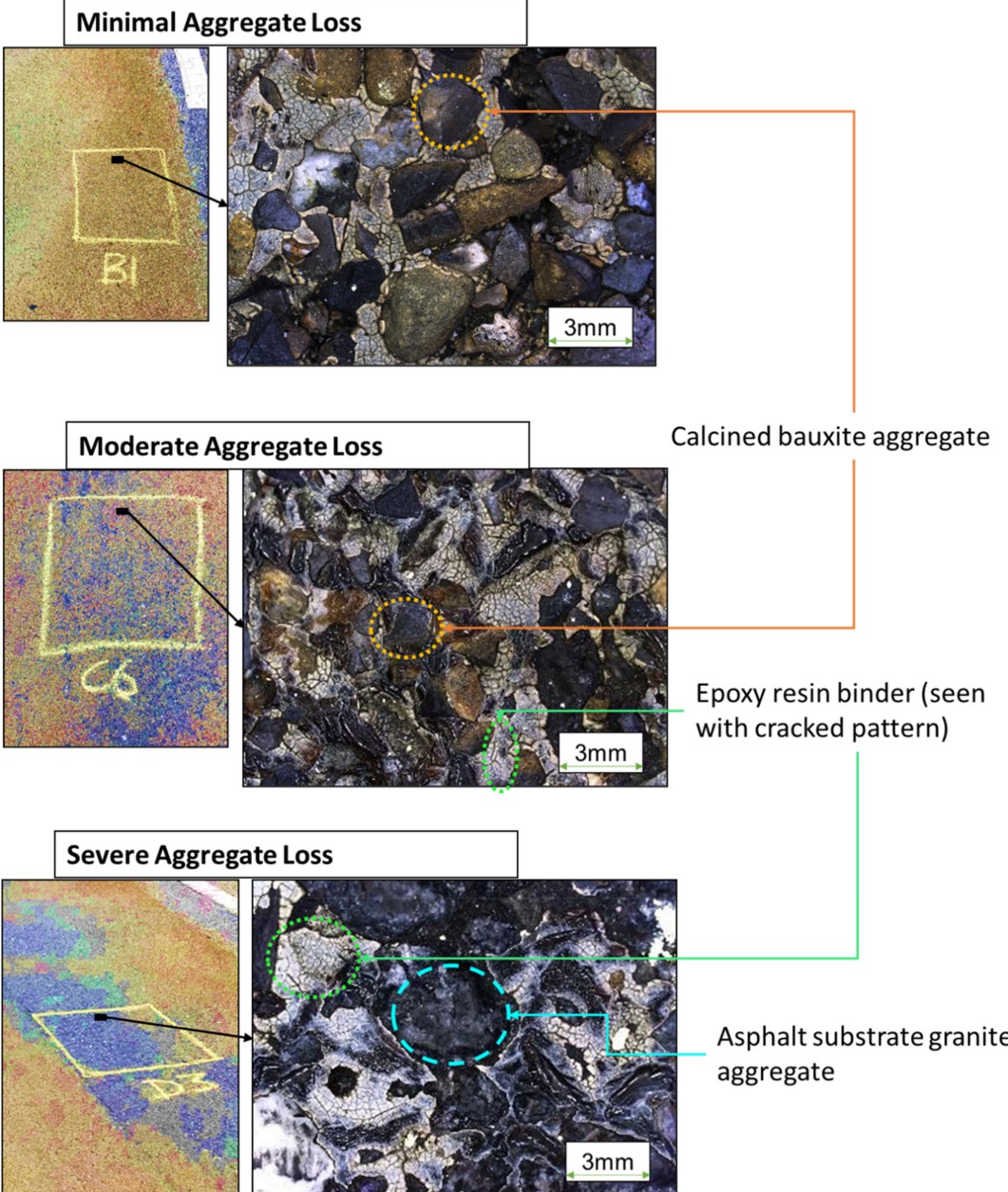

**Figure 12.** Photos showing example of three different aggregate loss severity types and microscopic image of the corresponding severity type. The photos (left) are enhanced with 400% saturation to distinguish calcined bauxite aggregates (represented as yellow) from epoxy and background asphalt aggregates (represented as green and blue).

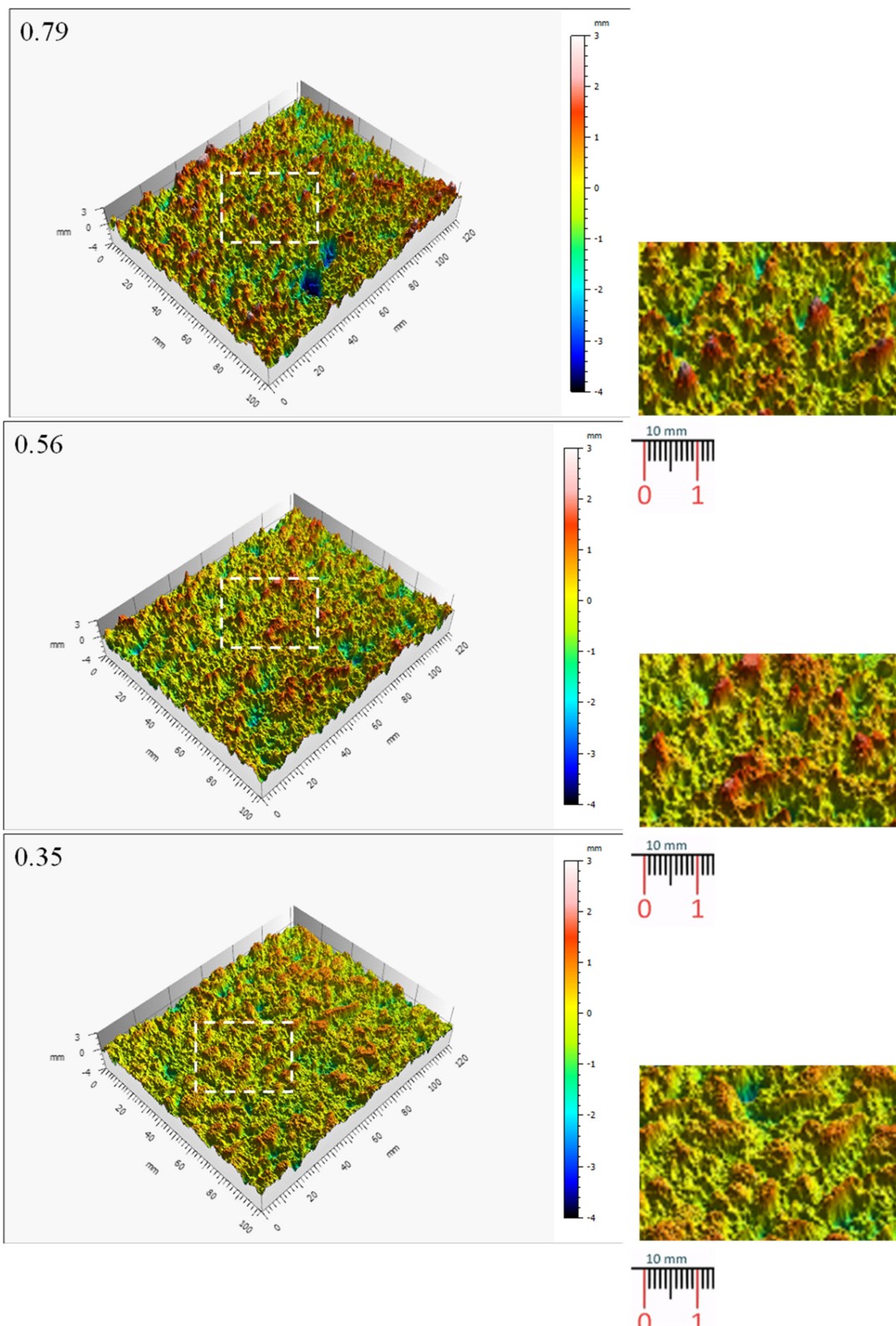

**Figure 13.** Isometric views of the topographical maps for three deterioration levels (from top to bottom: minimal with a coefficient of friction of 0.79; moderate 0.56; severe 0.35).

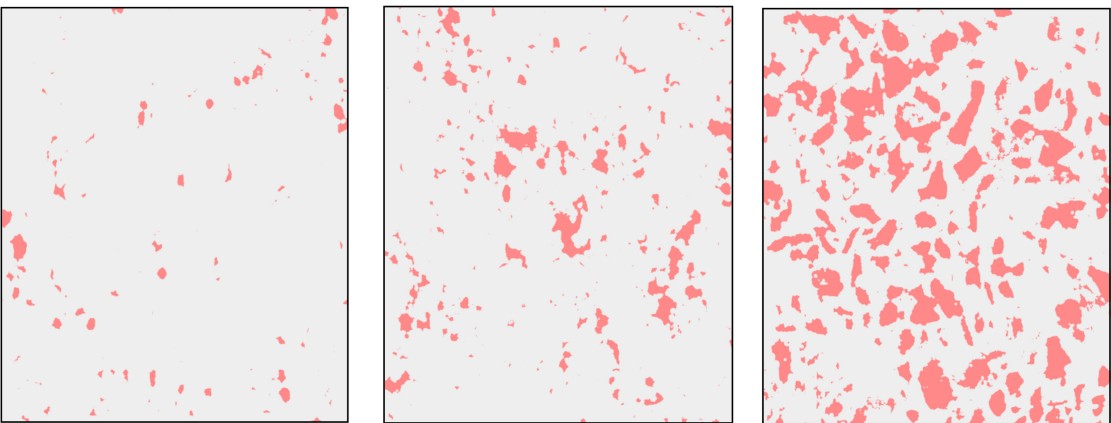

**Figure 14.** Projected areas for three aggregate loss levels (from left to right: minimal with a coefficient of friction of 0.79; moderate 0.56; severe 0.35).

### 3.2. Relationship between Texture Parameters and Coefficient of Friction

The relationship between the chosen texture parameters and the DFT friction coefficient, shown in Figures 15–17, corresponds to the three categories of parameters expressing the aggregates' characteristics, distribution, and aggregate loss physical process, respectively. Surfaces corresponding to three aggregate loss severity levels (minimal, moderate, and severe) are represented by triangles, squares, and circles, respectively.

Figure 15 shows the relationship between the aggregate characteristics and the DFT60 friction coefficient. It is seen that the higher the friction, the higher and sharper are the asperities. Therefore, the variation of the asperities' height (Sq) and angularity (Sdq and Ssc) with the friction coefficient reflects observations made from Figure 13.

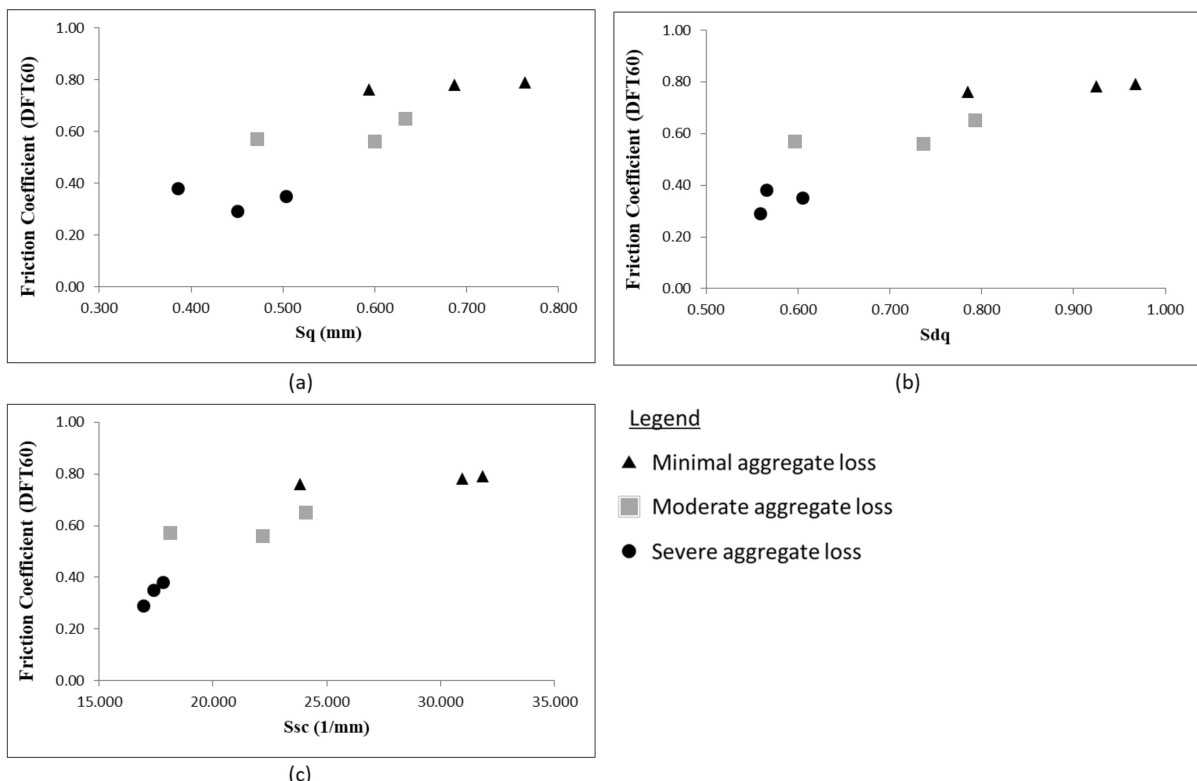

**Figure 15.** Relationship between DFT60 friction coefficient and parameters related to the aggregate characteristics including (**a**) root mean square height ($S_q$), (**b**) mean quadratic slope ($S_{dq}$), and (**c**) curvature ($S_{sc}$).

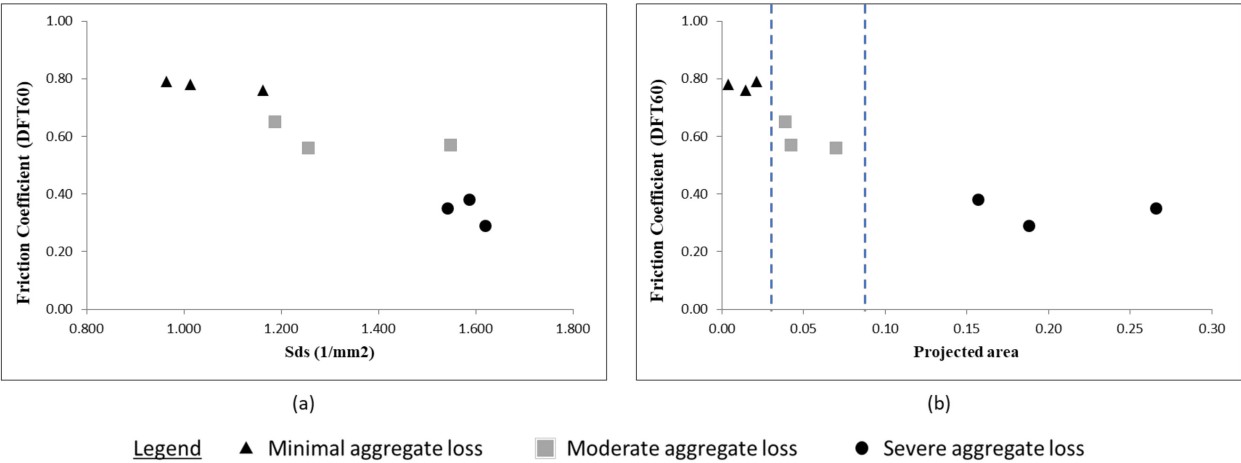

**Figure 16.** Relationship between DFT60 friction coefficient and parameters related to the aggregate distribution including (**a**) density ($S_{ds}$) and (**b**) projected area.

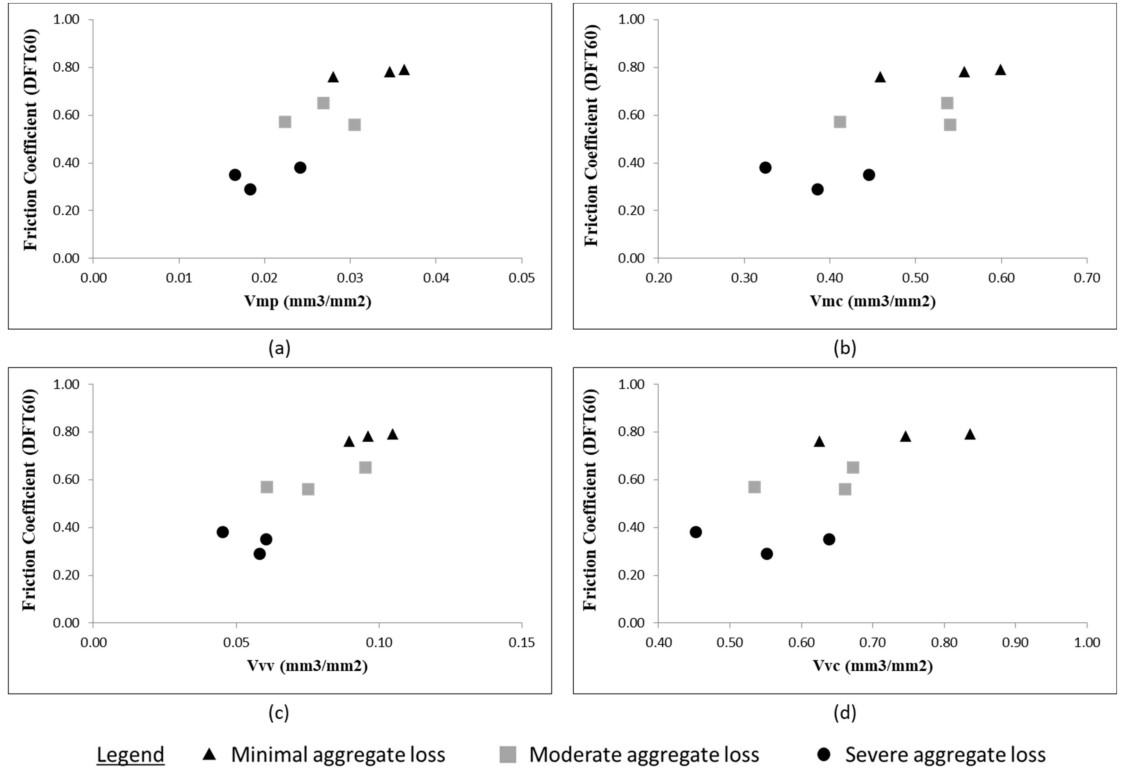

**Figure 17.** Relationship between DFT60 friction coefficient and volume parameters including (**a**) peak material volume (V_mp), (**b**) core material volume (V_mc), (**c**) valley void volume (V_vv), and (**d**) core void volume (V_vc).

Figure 16 shows the relationship between the aggregate distribution and the DFT60 friction coefficient. Again, the increase of the asperities' density and projected area reflects the observations made from Figure 14. Furthermore, it seems that the projected area allows separating the three aggregate loss levels (the dotted lines are placed arbitrarily to illustrate the separation); however, the test population is too small, so further validation is needed.

Figure 17 shows the relationship between the volume parameters and the DFT60 friction coefficient. The trend obtained with material volumes (Vmp and Vmc, corresponding to material peaks and material core) reflects the fact that as the HFS aggregate loss progresses (decrease of friction), there are less matter that can be removed by wear; it is then consistent with the progressive loss of aggregates. The decrease of the void vol-

umes for increasing deterioration expresses the flattening of the HFS surface following the material removal.

Table 1 provides a summary of one-way analysis of variance (ANOVA) results for nine texture parameters discussed in this section. The purpose of this analysis is to test whether there is a statistically significant difference in texture parameter mean values among the three aggregate loss severity levels (minimal, moderate, and severe aggregate loss). From Table 1, the *p*-values for majority of the texture parameters (except Vmc and Vvc) are smaller than the 0.05 significance level, which reveals that there is a statistically significant difference in seven texture parameter mean values between at least two aggregate loss severity levels. In addition, based on the relative high value of the F-value, the density (Sds), average asperity slope (Sdq), and average asperity curvature (Ssc) texture parameters can potentially be the best distinguishers of the different aggregate loss severity levels. Additional post-hoc tests need to be performed to confirm the significant differences between the individual aggregate loss severity levels for each texture parameter. Moreover, the current analysis uses limited data points at each aggregate loss severity level and hence, additional repetitions are required to increase the reliability of results.

**Table 1.** Results of one-way ANOVA for different texture parameters.

| Texture Parameter | *p*-Value | F-Value |
|:---:|:---:|:---:|
| Sq | 0.028 * | 6.891 |
| Sdq | 0.009 * | 11.372 |
| Ssc | 0.011 * | 10.635 |
| Sds | 0.006 * | 13.189 |
| Projected area | 0.031 * | 6.569 |
| Vmp | 0.022 * | 7.738 |
| Vmc | 0.079 | 3.987 |
| Vvv | 0.014 * | 9.355 |
| Vvc | 0.117 | 3.131 |

* The mean difference is significant at 0.05 level.

## 4. Discussions

As expected, texture parameters provide additional inputs to better understand the friction behavior of HFS. The correlation between the aggregates' characteristics (Sq, Sdq, and Ssc) and the coefficient of friction, besides the fact that it confirms the well-known effect of surface asperities' height and shape on friction, shows that HFS friction is primarily controlled by aggregates. Moreover, decreasing the material volume (Vmp and Vmc) with larger aggregate loss (and subsequent lowering of friction) also confirms the aggregate loss process. Results related to the aggregate distribution are more surprising. As described in Figure 4, one could expect that the number of contact points decreases as the HFS deteriorates (followed by a friction decrease); yet the density shows the opposite trend. A tentative explanation would be that these density parameters do not consider the nature of the asperities in contact with the tire; when the HFS is still sound (minimal level of aggregate loss), the asperities calculated by the software would be the calcined bauxite aggregates and, due to their sharp edges and high peaks, would induce a low density. However, with increasing aggregate loss, the software would detect asperities, which might be a mix between calcined bauxite aggregates (high peaks) and those from the underlying pavement (low peaks). This issue is related to how asperity is defined in the data extraction software and can be resolved in the future by providing better asperity definition customized to calcined bauxite characteristics. Similarly, results related to void volume are also unusual, decreasing void volume (especially Vvv) with increasing aggregate loss. One would expect the opposite trend due to the detachment of aggregates creating more voids in the surface. It is possible the voids are filled up with wear debris (dislodged aggregates, rubber material, dust) and circulation of wear debris at the tire/HFS interface, which should be considered in the future study of HFS deterioration.

To illustrate the relevance of texture parameters, they are plotted against an artificial timeline corresponding to friction deterioration observed at the NCAT test track. Four points are chosen in the friction timeline (from Figure 2) and corresponding texture parameter values are plotted on the friction timeline as shown in Figure 18. Sq and Vmc are chosen for illustration based on their high correlation with friction. From the graphs, we can find the two texture parameters mimic the friction deterioration, and the two parameters show a sudden deterioration corresponding to friction deterioration.

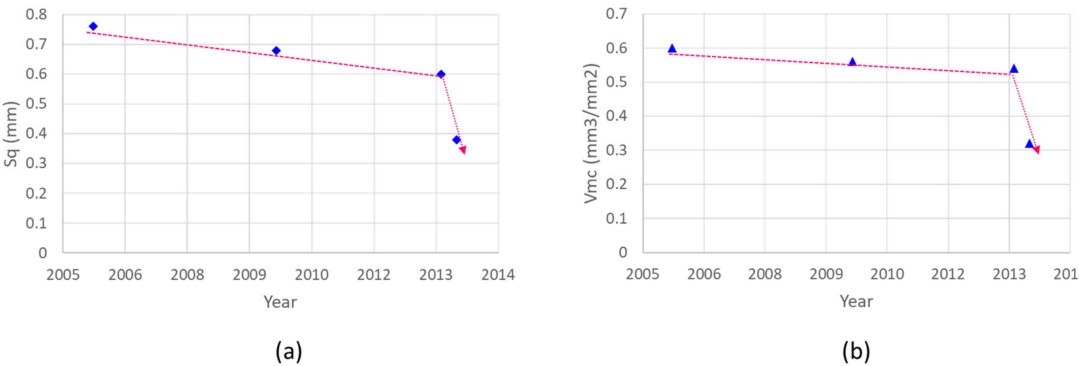

(a)

(b)

**Figure 18.** Plot of (**a**) Sq and (**b**) Vmc texture parameter values based on an artificial timeline corresponding to friction deterioration.

The discussion above highlights the relevance of the chosen texture parameters because, on the one hand, they confirm some intuitive results (effect of aggregates' height and shape on friction) and, on the other hand, they provide new insights into the contact conditions between the tire and the HFS (based on the projected area). Nevertheless, this investigation shows that the conducted analyses are not enough, especially with limited data points and methods used, to establish the HFS aggregate loss relationship (linear, logarithmic, s-shaped, etc.) with friction and to fully understand the HFS surface characteristics during the deterioration process. This investigation can benefit from a more diverse and larger dataset, and it can explore more complete analysis methods, like particles or multiscale analysis, to detect individual aggregates and group them into classes based on their characteristics (height, size, shape, etc.); this classification would help, in turn, determine the amount and the nature of aggregates that are removed as the deterioration process progresses.

It can also be seen that the current description of HFS deterioration (as a consequence of aggregates loss) is too simplistic because it does not consider other connected mechanisms like the transformation of dislodged aggregates and their circulation as a third body at the tire/road interface, etc. Although certain texture parameters are promising to be used as performance measures based on the correlations, correlations alone are not sufficient to establish the trigger point for taking safety and maintenance action based on the texture parameters. To go further, it would be necessary to adopt a more tribology-based approach to model the friction deterioration. One possibility is to reformulate the problem as the wear of a coated system.

HFS can be considered as a coating due to its protective function. The main difference between "usual" coatings and HFS is that the first is used to reduce friction and the second aims at increasing friction. Coatings have the form of a layer with inclusions as reinforcement, whose dimensions are much smaller than the thickness of the coating; HFS has a layer of epoxy whose thickness is lower than the size of the inserted aggregates. The usual contact condition of a coated system is either a hard substrate covered by a soft coating, or a soft substrate covered by a hard coating [23]; For HFS, the slider (the tire) is soft, and the coating and the substrate are hard. Accordingly, the contact between a tire and a HFS coating would be close to the one studied by Halling in [24]. This author considered

the friction and wear behavior of a system formed by two rough surfaces (numbered as 1 and 3 in Figure 19a), one of which is covered by a coating layer (Number 2 in Figure 19a).

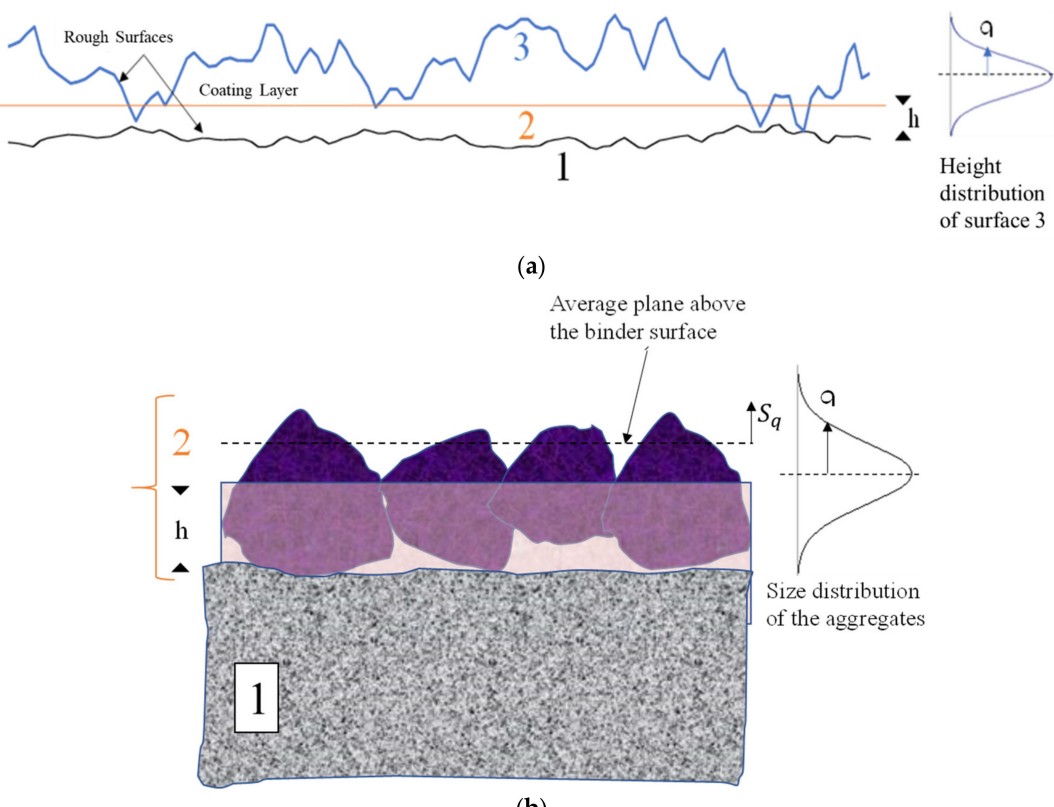

**Figure 19.** Geometry of the contact and related parameters (**a**) from Halling's work [22] and (**b**) adapted to tire/HFS contact (Coating 2 includes the binder and the aggregates).

A friction model, based on a distribution of friction between Surface 3/Coating 2 contact and Surfaces 3/1 contact, leads to a variation of the coefficient of friction resulting from this contact as illustrated in Figure 20a.

On the $x$ axis, h is the thickness of the coating layer and $\sigma$ is the standard deviation of the height distribution of Surface 3. The key idea is the trend, which is similar to a Stribeck curve in lubricated contacts: (1) friction is stable until a critical value of h/$\sigma$ is reached; (2) one observes a sudden decrease of friction to a minimum before (3) a slight increase for high values of h/$\sigma$. While Stage 3 is not applicable to the present study, Stages 1 and 2 would reflect the change of HFS friction.

To adapt Halling's geometry of contact, one can assume the following: Surface 1 is that of the underlying pavement; Surface 3 is that of the tire and can be considered, as a first approximation, as smooth; and Surface 2 is that of the HFS with h is the thickness of the epoxy binder and $\sigma$ is the standard deviation of the height distribution of the calcined bauxite aggregates (Figure 19b). Considering an embedment of the aggregates of 50% (which is a good application of the HFS [7]), a first approximation of $\sigma$ could be $\sigma = 2Sq$ (Figure 19b). With increasing wear, e.g., increasing loss of aggregates and then decreasing Sq (based on Figure 15a), the ratio h/$\sigma$ should increase from 0.5 (initial application) to 1 when the tire touches the binder. To plot the graph in Figure 20b, it was assumed that h/$\sigma$ = 0.5 for the minimal aggregate loss (close to the surface state just after the application); an adjusted value of h = 0.7 mm. A thicker layer (h = 1.5 mm) would be more representative, considering the usual size of the aggregates employed for HFS (3–4 mm after [3]). It should be noted that h could be higher if one considers maximum height parameter from the

topographical maps. However, this parameter was not used because of its sensitivity to extreme height values.

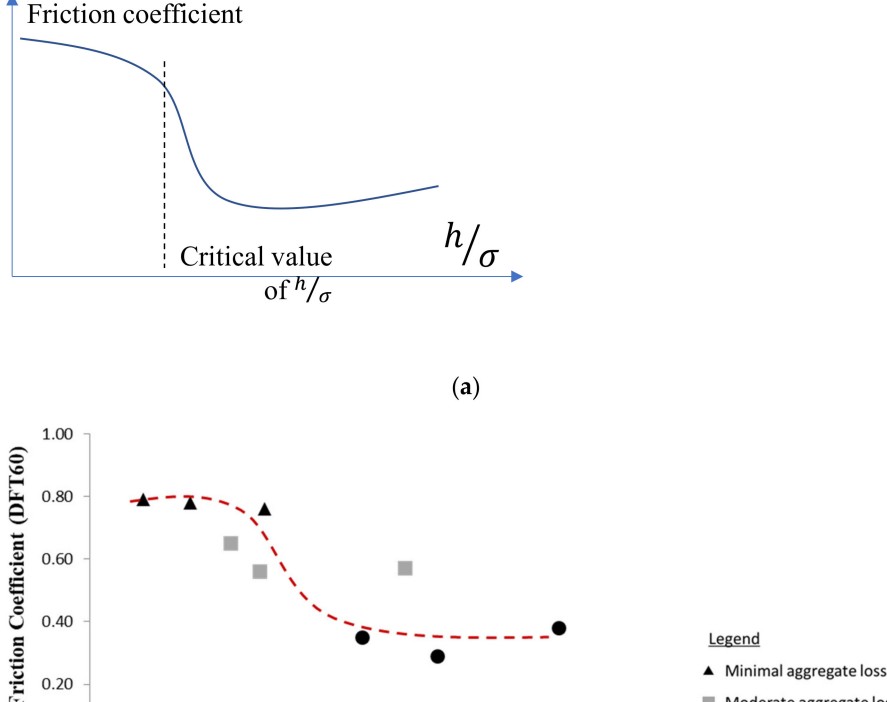

(**a**)

(**b**)

**Figure 20.** Variation of the coefficient of friction with the ratio h/σ (**a**) trend from Halling's work [22] and (**b**) data from the present study.

Figure 20b shows a decrease of the coefficient of friction when h/σ increases. A global linear variation can be seen, but one can also expect a S-shaped variation as illustrated by the dotted line drawn "by eye." Obviously, more data and a more complete modeling are needed to confirm the S-shaped variation. Nevertheless, the coating-inspired approach would provide a way to explain the sudden friction drop (due to a change of "lubrication" regimes) and predict this drop.

To confirm the relevance of this approach, tests under controlled conditions would be conducted to follow the evolution of an HFS surface when it is subjected to traffic-induced wear. Laboratory tests are necessary to simulate tire passes, and, besides collecting friction and texture measurements, the tire-HFS interface needs to be visualized. This visualization, not applicable to field tests, is essential because it can provide more insights into the involved phenomena. From this visualization, an analysis of the flows of debris would help to better understand the lubrication regimes at the tire-HFS interface and provide a physical interpretation of the graph in Figure 20b and a modeling of the observed trend. Other factors like the roughness of the substrate (underlying pavement) deserve more attention, as well, because the substrate's asperities can induce high stress on the coating [25] and, by cracking, accelerate its damage and the debonding of the aggregates.

## 5. Conclusions and Recommendations

In this work, a texture analysis was conducted to characterize the variation of high-friction surfaces (HFS) over time and interpret their friction deterioration. Based on the main failure mechanism (aggregate loss) observed on test sections at the National Center

for Asphalt Technology (NCAT) Test Track, texture parameters were chosen to characterize the aggregates' height (root-mean-square Sq) and shape (slope Sdq and curvature Ssc), the aggregates' distribution (density and projected area) and the aggregate loss process (volume parameters). The chosen texture parameters were correlated with friction. Preliminary findings from the correlation analysis and ANOVA reveal (a) texture parameters related to asperities' height, shape, and material volume show reasonable correlation with friction, (b) texture parameters related to density, projected area, and void volume also show correlation, but the trend is contrary to what is expected, and (c) density (Sds), average asperity slope (Sdq), and average asperity curvature (Ssc) are promising texture parameters to distinguish the different aggregate loss severity levels. Since, texture parameters (including height, shape, material volume) have shown reasonable correlations with friction and are able to reasonably distinguish different aggregate loss severity levels, it indicates they can be potentially used as alternate performance measures to monitor HFS deterioration (supplement traditional friction measurements).

In addition, it was deemed necessary to adopt a more tribology-based approach to model the HFS deterioration to predict its friction deterioration and establish trigger criteria for HFS maintenance. The aggregate loss phenomenon and associated friction deterioration has been reformulated considering wear of a coated system (HFS can be treated as a coating that protects the underlying pavement). A novel approach is proposed to detect the HFS friction coefficient transition based on aggregate loss by adapting the contact conditions similar to the change of lubrication regimes in a lubricated contact. Based on a limited number of test surfaces (9), it was possible to observe a decrease of the coefficient of friction when $h/\sigma$ increases similar to an S-shaped Stribeck curve. The sudden drop of friction when $h/\sigma$ is above a critical value opens the possibility of better understanding the HFS deterioration process and predicting the transition from mild to rapid friction decrease.

Recommendations for future works include the following:

(1) Perform additional testing with a larger and more diverse data set to establish the actual relationship (linear, logarithmic, s-shaped, etc.) between texture parameters and friction deterioration.

(2) Refine the definition of asperity for computing the density parameter to improve the results by considering the calcined bauxite aggregate's typical size (3–4 mm) and asperity's contact conditions with a rubber tire/DFT slider. For example, asperities with heights and widths greater than 1.5 mm must only be included in the density computation.

(3) Explore methods to detect individual aggregates and statistically analyze their characteristics (height, shape, density) to better understand the HFS aggregate loss deterioration process.

(4) Perform additional tests under controlled conditions to confirm the relationship between friction coefficient and $h/\sigma$ value.

(5) Although the evolution of aggregate loss was not monitored over time in this study, it would be ideal to perform long-term monitoring of aggregate loss and conduct similar analysis presented in this paper to gain much better understanding of HFS deterioration.

(6) Explore other 3D pavement scanning devices and texture measurement methods to calculate the identified texture parameters and evaluate their feasibility to detect HFS deterioration.

**Author Contributions:** Conceptualization, C.P., M.-T.D. and Y.-C.T.; methodology, C.P., M.-T.D. and Y.-C.T.; software, C.P. and M.-T.D.; validation, C.P., M.-T.D. and Y.-C.T.; formal analysis, C.P. and M.-T.D.; investigation, C.P. and M.-T.D.; resources, C.P., M.-T.D. and Y.-C.T.; data curation, C.P.; visualization, C.P. and M.-T.D.; writing—original draft preparation, C.P. and M.-T.D.; writing—review and editing, C.P., M.-T.D. and Y.-C.T. All authors have read and agreed to the published version of the manuscript.

**Funding:** This research received no external funding.

**Institutional Review Board Statement:** Not applicable.

**Informed Consent Statement:** Not applicable.

**Data Availability Statement:** The data presented in this study are available on request from the corresponding author.

**Acknowledgments:** The authors would like to thank the engineers and researchers at NCAT including Micheal Heitzman, Buzz Powell, Jason Nelson and Vickie Adams for their inputs and providing the assistance in the data collection.

**Conflicts of Interest:** The authors declare no conflict of interest.

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
