# Peer review of "Analysis of High-Friction Surface Texture with Respect to Friction and Wear"

_coatings, doi:10.3390/coatings11070758_

Round 1

Reviewer 1 Report

Dear authors,

Paper refers to the characterization of surface texture and friction coefficient evaluation. I suggest you add the following contents to the article

  1. Please expand the current state-of-the-art. Describe the methods currently used to evaluate road surface texture and how your approach is better than others. The authors should clearly indicate what new elements the methodology has been supplemented with or what new quality has been added.  
  2.  Explain why you chose these surface texture parameters. Moreover, the authors used the parameter which characterizes the height difference between the highest peak and deepest valley in the profile but previously removed the extreme values. This may have affected the results. Explain
  3. Expand the article to include evaluation using other parameters such as volume parameters. Perhaps other parameters will achieve better/higher values of the coefficient of friction and interesting conclusions. You can do this in the supplementary materials
  4. Nowadays, a multiscale approach is used for surface evaluation, such as road surfaces. Try to take this into account in your research (it is also available in Mountains software). This will allow to determine the size and amount of surface aggregates and particles that are removed during the life of the road.  This adds value to your study. It should be also accompanied with statistical analysis what now is missing
  5. Isometric views of the post-measurement areas are missing.
  6. The paper analyzes 9 spots. This is not a very large number of samples, it would be worth to consider to increase the number of samples. In Figures 14 and 15 it is worth to mark which dots correspond to minimal, moderate and severe aggregate loss
  7. The conclusion should be improved, extended

Kind regards
Reviewer

Author Response

We thank the reviewer for providing valuable comments which have helped us improve our paper. Please find below the response to each comment. 

1. Please expand the current state-of-the-art. Describe the methods currently used to evaluate road surface texture and how your approach is better than others. The authors should clearly indicate what new elements the methodology has been supplemented with or what new quality has been added.  

There is a misunderstanding in the purpose of the paper (the revised version should clarify this point). We would like to investigate the potential of texture parameters to better understand the aggregate loss process and predict HFST friction deterioration before damages are visible. In a first step, the priority is to identify texture parameters that can be used as alternate performance measures   to detect HFS deterioration (rather than just using friction). The second step is to research the monitoring devices that collect data and calculate the identified parameters. In this paper, our focus is on the first step. Exploring state-of-the-art of existing texture measuring methods is valuable but out of the scope of the paper.

2. Explain why you chose these surface texture parameters. Moreover, the authors used the parameter which characterizes the height difference between the highest peak and deepest valley in the profile but previously removed the extreme values. This may have affected the results. Explain

A “texture parameter characterization” section has been added to justify the choice of texture parameters. The RVD parameter that characterizes the height difference between the highest peak and deepest valley in the profile is no longer used.

3. Expand the article to include evaluation using other parameters such as volume parameters. Perhaps other parameters will achieve better/higher values of the coefficient of friction and interesting conclusions. You can do this in the supplementary materials

We have added volume parameters and projected area, which reflect the contact area between the aggregates and the tires.  

4. Nowadays, a multiscale approach is used for surface evaluation, such as road surfaces. Try to take this into account in your research (it is also available in Mountains software). This will allow to determine the size and amount of surface aggregates and particles that are removed during the life of the road.  This adds value to your study. It should be also accompanied with statistical analysis what now is missing

The analysis using the multiscale approach (using the “grains and particles” module in the MountainsMap software), would deserve a separate paper and your valuable suggestions are cited in the conclusions as a perspective for future research.

5. Isometric views of the post-measurement areas are missing.

Isometric views of surfaces representing three friction levels have been added.

6. The paper analyzes 9 spots. This is not a very large number of samples, it would be worth to consider to increase the number of samples. In Figures 14 and 15 it is worth to mark which dots correspond to minimal, moderate and severe aggregate loss

We agree that the number of surfaces in the paper is limited. The small population is due to the fact that it is difficult to find areas with different deterioration levels on actual roads. Nevertheless, we have tried to select samples with distinct friction levels. Per your suggestion, we have included different markers corresponding to minimal, moderate, and severe aggregate loss.

7. The conclusion should be improved, extended

The conclusion has been rewritten with perspective for future research inspired from coated systems.

Reviewer 2 Report

- The authors have made a great research effort and collected valuable experimental data on the performance of HFST.

- I highly recommend the authors to revise the paper in terms of its grammar and typesetting and make sure it doesn’t have grammatical errors (examples: don’t use informal phrasal verbs such as “watch out”; Figure 10 “green and blue color are predominant”).

- The authors state that they are investigating the deterioration of HFST at the “end of its life”, and that service life is 7 to 12 years based on the paper. So, this is quite normal to see a product coming to the end of its service life, and we can replace it with a new one. From that perspective, what is the need for performing this research in the first place?

- The second sentence of the second paragraph of Introduction needs to be rewritten.

- If possible, change the text in Fig. 11 to English.

- Most of the observations explained in “discussions” section seem to be intuitive (probably except for the relationship between density and DFT). What is the benefit of performing these sophisticated measurements?

- Do you think this approach (collecting LS-40 or CTM data) is applicable for large areas? How do you explain the practical significance of this study?

- The “conclusions” section does not adequately present the research findings. More than half of this section is about future work, which is appreciated, but the focus should be on summarizing the results/conclusions first. 

Author Response

We thank the reviewer for their valuable comments which were helpful in improving the paper. Please see below the responses to individual comments.

-The authors have made a great research effort and collected valuable experimental data on the performance of HFST.

We thank the reviewer for acknowledging our effort.

- I highly recommend the authors to revise the paper in terms of its grammar and typesetting and make sure it doesn’t have grammatical errors (examples: don’t use informal phrasal verbs such as “watch out”; Figure 10 “green and blue color are predominant”).

Thank you for your suggestion.  We have revised the paper with help of a professional editor to improve the grammar and typesetting.

- The authors state that they are investigating the deterioration of HFST at the “end of its life”, and that service life is 7 to 12 years based on the paper. So, this is quite normal to see a product coming to the end of its service life, and we can replace it with a new one. From that perspective, what is the need for performing this research in the first place?

The HFS friction can transition from a “mild” to a “rapid” decrease of friction, which indicates the performance-based end of life. Moreover, the knowledge about the potential end of life is important for road authorities because they can schedule works to replace the damaged road. To fully understand the behavior of HFS, it is necessary to follow road sections from their installation to their end of life. Test tracks like NCAT facilities offer this possibility, and this is the reason why the words “end of life” have been used in the text. We agree that this wording can induce confusion about the aim of the research. The related sentences have been rewritten.

- The second sentence of the second paragraph of Introduction needs to be rewritten.

The introduction has been revised.

- If possible, change the text in Fig. 11 to English.

All graphs/views from the MountainsMapÒ software now include legends in English.

- Most of the observations explained in “discussions” section seem to be intuitive (probably except for the relationship between density and DFT). What is the benefit of performing these sophisticated measurements?

These sophisticated measurements helped confirm the impact of aggregate loss on the HFS friction deterioration. The study has provided new insights into the failure process of high-friction surfaces; now, we do not have to rely on the sole knowledge of friction.  In addition, an extended discussion has been added with reference to works on the tribological behavior of coated systems. The measurements performed help identify other approaches that could help predict sudden drops of friction and, thereby,  prevent accidents.

- Do you think this approach (collecting LS-40 or CTM data) is applicable for large areas? How do you explain the practical significance of this study?

The LS-40 has been used because it provides high-resolution data to analyze the involved phenomena and identify the texture parameters associated with these phenomena. These findings would provide relevant inputs for monitoring devices to define a suitable strategy in terms of data collection and analysis. While further tests are needed, the results show that texture parameters can be used to detect poor (low friction) HFST friction performance, and this research opens new perspectives on the use of texture measurement and leverages 3D pavement scanning technologies (that state DOTs in the U.S. have  already been used to collect full lane 3D pavement data)   to cost-effectively and regularly monitor HFST friction deterioration.

- The “conclusions” section does not adequately present the research findings. More than half of this section is about future work, which is appreciated, but the focus should be on summarizing the results/conclusions first. 

The conclusion has been rewritten with reference to new findings.

Reviewer 3 Report

The conclusions are more likely discussion. Can you please write some bulleted conclusions? The background and recent studies are not well explored. There are some studies. The recent textbooks are also a source. I found a recent publication, pavement design - materials, analysis and highways. Some recent studies are recommended. 

Author Response

We thank the reviewer for their valuable comments and these comments have been helpful in improving our paper. 

The conclusions are more likely discussion. Can you please write some bulleted conclusions? The background and recent studies are not well explored. There are some studies. The recent textbooks are also a source. I found a recent publication, pavement design - materials, analysis and highways. Some recent studies are recommended. 

Response:

The conclusion has been rewritten with reference to new findings.

For the state-of-the-art, we have added references to works conducted elsewhere; in particular we referenced recent works (from 2019) in the UK and the US.

Reviewer 4 Report

In general, this paper was written in haste. It has many basic errors and appears to be more a technical report. The importance and novelty of this work is low. Here are my detailed comments:

Introduction:

  1. First paragraph describes situation in the USA. Brief summary for other regions is highly appreciated.
  2. Second paragraph is focused on the outcome of National Center for Asphalt Technology. Are there other research center around the world that deal with the road friction. I suppose there are many people involved in similar research (e.g. SAE - https://doi.org/10.4271/2000-01-1314) what is somehow neglected here. In other words, this paper requires an in-depth analysis of the literature to indicate the missing gaps in the field and provide strong rationale for your study.
  3. Measuring surface topography of road and other large scale objects is a difficult task and differentiates from fine scale (<10 micrometers) measurements. A short ovierview of methods and instrumentation should be presented in this section. In addition, research showing correlations between friction and surface topography should be briefly summarized.

Background

  1. Quality of figure 1 should be improved. All similar figures should be made in the professional software or appearing to look professional.
  2. Characterization of surface toprography is scale dependent. Naming one characteristics a form, roughness or waviness should be done carefully. Here is a recent paper which summarizes that aspect and also covers multiscale effects 10.1016/j.cirp.2018.06.001
  3. Figure 2 should be improved - it seems to scanned from other physical source.
  4. A brief introduction of the tribological/physical effects occurring in the analyzed phenomena of interest should be given. The nature of these tribological effects is quite complex and not covered here.

Experiments and set-up

  1. This part is shown very superficially so it is not possible to replicate any of the measurements. Simply showing figures 5 to 8 with very brief description does not make it possible. I would expect a detailed presentation of your methods.

Texture analysis

  1. What kind of data filtration was used. Were the profiles leveled before analysis. It seems that the software used for the analysis was MountainsMap which allows to conduct proper data processing and quantificiation. Figure 11 has a description in French.
  2. Operation as in Figure 12 is well-known and covered in the standard. I suppose this figures has been derived from other source without mentioning it. A violation copyrights is possible. Please check that!
  3.  Figure 13 should be recreated to provide same standard as in other figures.
  4. On what base equations in this part were derived?

Results

  1. Statistical analysis is introduced (R^2) but it was not mentioned before. What about proper discrimination analysis - ANOVA? p-value?
  2. This results do require proper and complex multicriteria statistical analysis what is missing.
  3. Quality of figure 15 is bad.

Discussion

This part is not a discussion as it comes directly from the results. It does not provide any substantial explanation on the nature of the phenomena nor discusses the important aspects of this study with a reference to other works.

Some minor issues:

1. Second author's affiliation is uknown.

Author Response

We thank the reviewer for their valuable comments and these comments have been helpful in improving our paper. Please find below our responses to individual comments. 

Introduction:

  • First paragraph describes situation in the USA. Brief summary for other regions is highly appreciated.

Response: Works conducted in the UK and New-Zealand have been added.

  • Second paragraph is focused on the outcome of National Center for Asphalt Technology. Are there other research center around the world that deal with the road friction. I suppose there are many people involved in similar research (e.g. SAE - https://doi.org/10.4271/2000-01-1314) what is somehow neglected here. In other words, this paper requires an in-depth analysis of the literature to indicate the missing gaps in the field and provide strong rationale for your study.

Response: There is a misunderstanding in the purpose of the paper (the revised version should clarify this point). We would like to investigate the aggregate loss process and predict HFST friction deterioration before damages are visible. To make this research possible, we needed a test site that makes possible (in terms of experimentations as well as available data) the survey of an HFST section  from its installation to its end of life. As experiments were conducted by Georgia Tech, the best choice was the NCAT test track.

Research conducted in the UK and New Zealand has been added. The background section has been improved to include results from these countries, as well as field observations from the US DOT. Missing gaps have been identified and justification of the conducted research has been made in a newly added “research methodology” section.

  • Measuring surface topography of road and other large-scale objects is a difficult task and differentiates from fine scale (<10 micrometers) measurements. A short ovierview of methods and instrumentation should be presented in this section. In addition, research showing correlations between friction and surface topography should be briefly summarized.

Response: We have focused the paper on the failure of high-friction surfaces. Methods to measure road surface topography and texture-friction relationship are vast topics; a review would be insufficient if it is too brief or out of the scope of the paper if it is exhaustive.  Therefore, the paper has been rewritten to highlight more aspects related to the tribology of coated systems.

Background

  • Quality of figure 1 should be improved. All similar figures should be made in the professional software or appearing to look professional.

Response: Figures have been refined to look professional.

  • Characterization of surface topography is scale dependent. Naming one characteristic a form, roughness or waviness should be done carefully. Here is a recent paper which summarizes that aspect and also covers multiscale effects 10.1016/j.cirp.2018.06.001

Response: To focus the discussions on the tribological aspects, we have kept only LS-40 data (3D maps). The texture parameters are calculated from the whole topographical maps with the pre-processing method indicated. No reference is made to roughness, waviness, or form scales.

  • Figure 2 should be improved - it seems to be scanned from other physical source.

Response: The figure has been replaced by an illustration of better quality.

  • A brief introduction of the tribological/physical effects occurring in the analyzed phenomena of interest should be given. The nature of these tribological effects is quite complex and not covered here.

Response: A brief interpretation of the tribological effect of aggregate loss is presented in the HFS failure mechanisms section. We agree the tribological effects are quite complex, and this study attempts to analyze these impacts.

Experiments and set-up

  • This part is shown very superficially so it is not possible to replicate any of the measurements. Simply showing figures 5 to 8 with very brief description does not make it possible. I would expect a detailed presentation of your methods.

Response: The description of the measurement devices has been enhanced (test protocol added). Together with section “Data collection procedure,” readers would be able to reproduce the tests.

Texture analysis

  • What kind of data filtration was used. Were the profiles leveled before analysis. It seems that the software used for the analysis was MountainsMap which allows to conduct proper data processing and quantificiation. Figure 11 has a description in French.

Response: Please refer to the first sentence of the section “Pre-processing”;  the software is actually MountainsMaps, and it was used to process all data and calculate all parameters. All 3D maps were levelled as indicated in  Step 3 of the pre-processing. All figures have been changed to display only English legends.

  • Operation as in Figure 12 is well-known and covered in the standard. I suppose this figures has been derived from other source without mentioning it. A violation copyrights is possible. Please check that!

Response: The figure has been replaced by an  illustration we have created.

  • Figure 13 should be recreated to provide same standard as in other figures.

Response: The figure has been  removed. A  definition of the RVD parameter is provided in  Figure 3.

  • On what base equations in this part were derived?

Response: The equations were extracted from the following reference (added as reference 21 in the paper):

Stout, K. J., Sullivan, P. J., Dong, W. P., Mainsah, E., Luo, N., Mathia, T., Zahouani, H. (1993) The Development of Methods for the Characterisation of Roughness in Three Dimensions, Publication n° EUR 15178EN of the Commission of the European Communities (available  at www.bookshop.europa.eu)

Results

  • Statistical analysis is introduced (R^2) but it was not mentioned before. What about proper discrimination analysis - ANOVA? p-value?

Response: With a small population (due to the difficulty of  obtaining different levels of aggregate loss), it was not possible to build a true factorial experiment design. Therefore, ANOVA was not conducted.

  • This results do require proper and complex multicriteria statistical analysis what is missing.

Response: In this paper, we are looking for texture parameters that could help to better describe the aggregate characteristics and the aggregate loss process. In addition to explanations about physical phenomena, the criterion used to assess the relevance of a parameter is its correlation with friction.

With a small population (due to the difficulty of  obtaining different levels of aggregate loss), we believe  that complex statistical analyses would be biased.

  • Quality of figure 15 is bad.

Response: The figure is suppressed. In the revised version, the RVD parameter is illustrated in  Figure 3.

Discussion

This part is not a discussion as it comes directly from the results. It does not provide any substantial explanation on the nature of the phenomena nor discusses the important aspects of this study with a reference to other works.

Response: An extended discussion has been added with reference to works on the tribological behavior of coated systems.

 Some minor issues:

  • Second author's affiliation is uknown.

Response: The University Gustave Eiffel was created in January 2020. More information is available at the university website  (https://www.univ-gustave-eiffel.fr/en/).

Round 2

Reviewer 3 Report

Thanks for revising the article.

Author Response

Your feedback is valuable. Thank you. 

Reviewer 4 Report

The paper has been improved, yet there are some key issues still need to be resolved.

If you show any results from come from measurements you should provide enough replications to conduct statistical analysis. Otherwise the conclusions would lack fidelity. Therefore, I strongly urge the authors to show p-value for ANOVA as you present three? different groups, e.g. in figures 15-17. These figures also lack legends - as I cannot say what different mark represent... The quality of those figures is also very low. Please remove thin horizontal lines and make them look more professional.

Author Response

Thank you for your feedback and we really appreciate your suggestions. 

If you show any results from come from measurements you should provide enough replications to conduct statistical analysis. Otherwise the conclusions would lack fidelity. Therefore, I strongly urge the authors to show p-value for ANOVA as you present three? different groups, e.g. in figures 15-17.

Response: As per the reviewer's suggestion, we have included a table (Table 1) with the ANOVA p-value for the different groups (3 aggregate loss severity levels) shown in figures 15-17. 

These figures also lack legends - as I cannot say what different mark represent... The quality of those figures is also very low. Please remove thin horizontal lines and make them look more professional.

Response: The quality of the figures are now improved based on the reviewer's suggestion - removed the thin horizontal lines, added a legend, increased the figure size.